# Large polarization-dependent exciton optical Stark effect in lead iodide perovskites

Ye Yang[1], Mengjin Yang[1], Kai Zhu[1], Justin C. Johnson[1], Joseph J. Berry[1], Jao van de Lagemaat[1] & Matthew C. Beard[1]

A strong interaction of a semiconductor with a below-bandgap laser pulse causes a blue-shift of the bandgap transition energy, known as the optical Stark effect. The energy shift persists only during the pulse duration with an instantaneous response time. The optical Stark effect has practical relevance for applications, including quantum information processing and communication, and passively mode-locked femtosecond lasers. Here we demonstrate that solution-processable lead-halide perovskites exhibit a large optical Stark effect that is easily resolved at room temperature resulting from the sharp excitonic feature near the bandedge. We also demonstrate that a polarized pump pulse selectively shifts one spin state producing a spin splitting of the degenerate excitonic states. Such selective spin manipulation is an important prerequisite for spintronic applications. Our result implies that such hybrid semiconductors may have great potential for optoelectronic applications beyond photovoltaics.

[1] Chemistry and Nanoscience Center, National Renewable Energy Laboratory, Golden, Colorado 80401, USA. Correspondence and requests for materials should be addressed to Y.Y. (email: ye.yang@nrel.gov) or to M.C.B. (email: matt.beard@nrel.gov).

The optical Stark effect (OSE) results from a coherent interaction between excitonic states and a non-resonant pump photon field, and is not associated with the photo-generation of real carriers or excitons[1]. The OSE results in a shifting of the exciton transition energy to higher energy in a semiconductor by optical pumping with photon energies that are below the bandgap (non-resonant)[2,3]. Because of the coherent nature, the OSE has many promising applications, for example, in quantum information processing and communication, serving as a potential ultrafast optical switch[3,4], source of quantum entanglement[5,6] and a method for spin manipulation[7–10]. The OSE can also be utilized to produce ultra-short laser pulses serving as a passive mode-locking saturable absorber[11–13].

To serve as a good OSE system, semiconductor candidates require quantum states that are strongly coupled to photons to enable effective optical control and they should have narrow transition bandwidths that can exhibit a high-contrast spectral modulation. Quantum-confined semiconductors such as quantum wells[2,14,15], quantum dots[7,10,16,17], two-dimensional semiconductor monolayers[18–20] and carbon nanotubes[21] have been intensively studied for OSE because of their strong exciton–photon interactions and discrete exciton transitions. Unfortunately, detecting OSE in bulk semiconductors requires extremely low temperatures to achieve a well-resolved exciton transition that is usually buried in continuum band transitions[14,22]. On the other hand, recent spectroscopic investigations of lead-halide perovskites have uncovered that the excitonic transitions in these semiconductors have the prerequisite conditions for an OSE system; narrow bandwidth and large transition oscillator strength[23–26].

Here we report the observation of polarization-selective OSE of bandedge excitonic states in methylammonium lead iodide (MAPbI$_3$) perovskite films at room temperature. The OSE is generated and detected by using transient absorption (TA) spectroscopy. Under similar pump intensity and detuning, the observed OSE-induced energy shift is nearly an order of magnitude larger than in previously reported quantum-confined systems[2,20,21] and is similar to that observed in monolayer WSe$_2$, a two-dimensional semiconductor[19] that is gaining increasing interest in spintronic investigations. Within this context a broader set of applications beyond photovoltaic can be envisioned for these solution-processed hybrid semiconductors.

## Results

### Optical Stark effect.
We are able to selectively dress the excitonic states within MAPbI$_3$ films by using circularly polarized photons from a femtosecond laser source that is tuned below the exciton transition energy, so that ideally no excitons/carriers are created. A low-intensity circularly polarized broadband probe pulse, with photon energies that span the exciton transition energy, confirms that the dressed excitonic states are shifted towards higher energy, and the shift depends linearly on the pump intensity. The corresponding energy shift ($\delta E$), applying a perturbation of the exciton–photon interaction in a two-level system, is given by[19,21,27]

$$\delta E = \frac{(\mu_{0X})^2 \langle F \rangle^2}{\Delta} \qquad (1)$$

where $\mu_{0X}$ is the transition dipole moment between the ground and a non-degenerate exciton state, $\langle F \rangle$ is the time-averaged electric field of excitation light and $\Delta$ is the detuning of the excitation energy from the exciton resonance ($\Delta = E_0 - \hbar\omega$). From equation (1), the energy shift, $\delta E$, is proportional to both the square of the transition dipole moment and the excitation intensity as $\langle F \rangle^2 = 2I_0/(\sqrt{\varepsilon}\varepsilon_0 c)$, where $I_0$ is light intensity, $\varepsilon$ is dielectric constant of the sample, and $\varepsilon_0$ and $c$ are the vacuum permittivity

and speed of light, respectively. Optical selection rules implicitly govern $\delta E$ through the transition dipole moment.

Excitonic states are primarily built from conduction and valence bandedge states, and for lead-iodide perovskites both bands are twofold degenerate with a total angular momentum number of 1/2 (ref. 28). The two lowest bright exciton states are denoted by the azimuthal quantum number of the total angular momentum $|\pm m_j\rangle$. Optical transitions from the ground state to the exciton states are allowed for $\Delta m_j = \pm 1$. At large detuning energies compared with the exciton-binding energy ($\Delta \gg R_{ex}$) the degenerate excitonic states can be treated as multiple independent two-level systems and equation (1) is still valid to predict the energy shift of each excitonic state[29] (see Supplementary Note 1 for a derivation of equation (1)).

The polarization selectivity of the OSE is the basis for its experimental demonstration, and our experiment is described in Fig. 1. Excitonic states with $m_j = 1$ can only couple to circularly polarized photons with spin magnetic quantum number ($m_s$) of $+1$, represented by $\sigma+$, to preserve angular momentum. Therefore, when a $\sigma+$ photon couples to a $m_j = 1$ excitonic state a new entangled state results with transition energy $E_0 + \delta E$. Since this new entangled state also carries total angular momentum with $m_j = 1$, it can only be detected by another $\sigma+$ photon that is tuned to $E_0 + \delta E$. If, in contrast, a counter-polarized probe photon ($\sigma-$) with $m_s$ of $-1$ is used, no difference in the energy level will be detected since it cannot induce exciton transitions with $m_j = 1$. For the same reason, a $\sigma+$ photon cannot couple with an exciton state of $m_j = -1$, leaving that transition unperturbed.

### Linear absorption.
The preparation of the MAPbI$_3$ films is described in the Methods section, and the film is characterized by X-ray diffraction (Supplementary Fig. 1), showing a highly crystalline perovskite structure in the tetragonal phase. A relatively large in-plane grain size (around 1–2 μm) is indicated by the scanning electron microscope image (Supplementary Fig. 2). The linear absorption spectrum of the sample (Fig. 2, red circles) is determined by measuring transmission and reflection (Supplementary Fig. 3). There is a sharp onset followed by a nearly constant absorption, and such behaviour can be attributed to Coulombic interactions between the electron and hole within

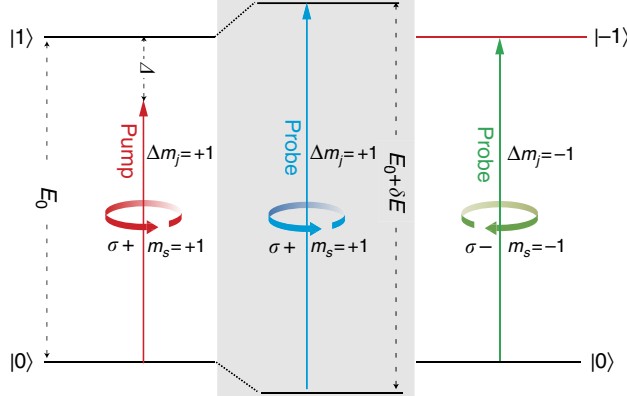

**Figure 1 | Energy diagram of the exciton energy shift due to OSE.** The diagram shows energy levels for the ground state, $|0\rangle$, and the two bright exciton states, $|\pm 1\rangle$, labelled by the azimuthal quantum number of the total angular momentum, $m_j$. The exciton transition energy is $E_0$. When a photon with energy $E_0 - \Delta$ and with spin momentum $\sigma+$ interacts with the $m_j = 1$ exciton, then an entangled state with energy $E_0 + \delta E$ forms (shaded box). The new state can be detected by a probe photon with the same spin, while the $m_j = -1$ exciton is left unperturbed.

the Elliott model (Fig. 2, black dash line)[23–25]. The absorption components of the excitonic and continuum transitions (green and blue dash lines, respectively) are decoupled from the spectral fitting. The exciton resonance ($E_0$) is centred at 1.631 eV, and the bandgap ($E_g$) of continuum transitions is found at 1.642 eV, which yields an exciton binding energy ($R_{ex} = E_g - E_0$) of 11 meV, consistent with recently reported values[25,30–32]. The full-width at half-maximum of the exciton transition is found to be 73 meV (fitting parameters are tabulated in Supplementary Table 1).

**Transient absorption.** A high-intensity polarized monochromatic pump pulse ($\hbar\omega_p = 1.55$ eV) is used to couple to the exciton states (thus the detuning, $\Delta$, is 81 meV and therefore $\Delta \gg R_{ex}$, justifying the use of equation (1)). The resulting change in absorption ($\Delta A$) is captured by a weak broadband polarized probe pulse ($\hbar\omega = 1.55$–2.1 eV). Because, in our experiment, OSE

is created and probed by femtosecond pulses, the duration of the OSE response is determined by the pump–probe correlation time (around 180 fs). There is a slight spectral overlap between the excitation pulse and the unperturbed exciton transition, resulting in a small population of excitons that quickly dissociate to form free carriers ($<5 \times 10^{16}$ cm$^{-3}$). Since the density of real carriers is too low to screen Coulomb interactions (Debye length is around 50 nm) the major effect is to interfere with the observation of the OSE through the reduction of the exciton transition strength via phase space filling. The carrier-induced TA signal lasts for over tens of nanoseconds until the carriers recombine[30], however, due to the drastic difference in lifetime, $\Delta A$ resulting from OSE and from the presence of carriers can be isolated from one another using a simple spectral subtraction procedure (Supplementary Figs 4 and 5). To focus on the OSE, we only show the subtracted TA spectra (raw TA spectra are provided in Supplementary Fig. 4).

The OSE is first studied under different configurations of pump–probe polarization. The pump pulse intensity ($I_0$) is controlled to be the same regardless of polarization in this experiment. When the pump and probe are co-circularly polarized, denoted as ($\sigma+,\sigma+$), a prominent absorption change anti-symmetric to the exciton resonance centre ($\hbar\omega = 1.631$ eV) appears at zero pump–probe delay (Fig. 3a). The narrow temporal width (around 200 fs) of $\Delta A$ is consistent with the pump–probe correlation time. On the contrary, when the pump and probe pulses are circularly polarized opposite to one another, denoted as ($\sigma+,\sigma-$), no OSE features are detected (Fig. 3b). The above observations suggest that the $\sigma+$ pump selectively induces an energy shift of the exciton transitions with $\Delta m_j = +1$, and the $\sigma+$ probe can detect the modified exciton transitions while the $\sigma-$ probe cannot see them, consistent with angular momentum conservation considerations discussed above. The pump–probe configurations of ($\sigma-,\sigma-$) and ($\sigma-,\sigma+$) show the same spectra as ($\sigma+,\sigma+$) and ($\sigma+,\sigma-$), respectively (Supplementary Fig. 6).

The sample is also examined under two linearly polarized pump–probe configurations, parallel (VV) and perpendicular (HV) pump–probe polarization. The TA spectra show that similar $\Delta A$ signals due to OSE are observed under both configurations, and the magnitudes are around half of that measured under ($\sigma+,\sigma+$) configuration (Supplementary Fig. 7).

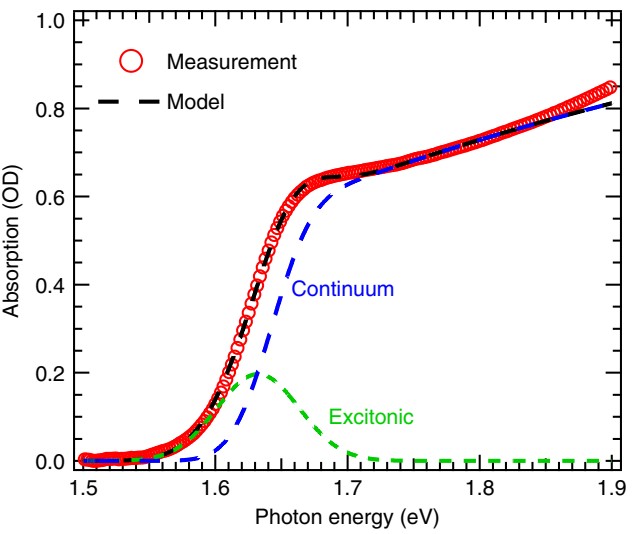

**Figure 2 | Absorption spectrum and Elliott's model of the MAPbI₃ films.** The absorption spectrum (red circles) near the bandgap was fit by Elliott's model (black dash line). The contribution from excitonic (green dash line) and continuum band (blue dash line) transitions are also plotted.

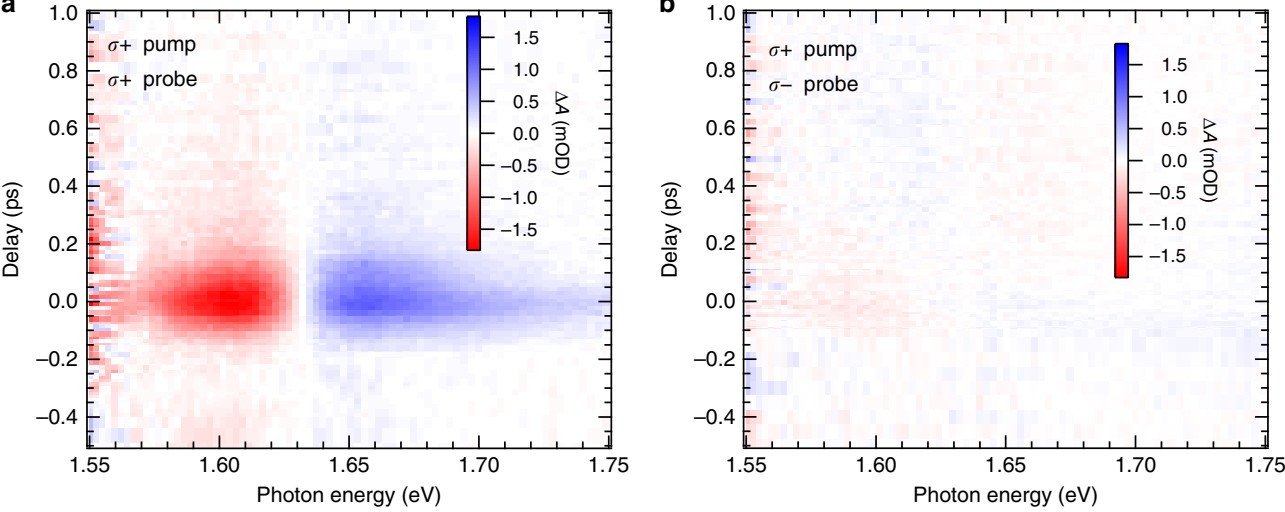

**Figure 3 | OSE-induced TA spectra of MAPbI₃ films.** The TA spectra are measured by (**a**) co- ($\sigma+,\sigma+$) and (**b**) counter-circularly ($\sigma+,\sigma-$) polarized pump–probe configurations. The y axis and x axis represent the pump–probe delay and probe photon energy, respectively. As indicated by the colour scale bars, the blue and red represent the photon-induced absorption and bleach, respectively. The colour intensity reflects signal magnitude.

Because linear polarization consists of equal amounts of $\sigma+$ and $\sigma-$ components, both (VV) and (HV) configurations can be considered as a mixture of 50% $(\sigma\pm,\sigma\mp)$ and 50% $(\sigma\pm,\sigma\pm)$. Therefore, the OSE magnitude detected by the (VV) and (HV) configurations are the same, and the magnitude is only 50% of that detected by $(\sigma+,\sigma+)$.

## Discussion

Because we use TA to observe the OSE, the resulting $\Delta A$ spectra arise from the energy shift of the exciton transition, which are the difference spectra of shifted and original exciton absorption. Since $\delta E$ is much smaller than $E_0$, $\Delta A$ can be approximated by the following equation:

$$\Delta A = -\frac{1}{2}\frac{\partial A_{ex}}{\partial \hbar\omega}\cdot\delta E \qquad (2)$$

where $\hbar\omega$ is probe photon energy and $A_{ex}$ is the exciton absorption spectrum obtained from deconvolution of the linear absorption spectrum (Fig. 2). The pre-factor of 1/2 accounts for the selection rule of $(\sigma+,\sigma+)$ configurations because only half of the exciton states are coupled to the $\sigma+$ pump and $\sigma+$ probe each. We also investigated the dependence of the OSE on $I_0$ that represents the photon energy flux within the pulse duration. The $\Delta A$ recorded under different values of $I_0$ are plotted in Fig. 4a (red circles), the blue traces are non-linear least squares best-fits of equation (2) to the data with $\delta E$ as the only adjustable parameter. The model deviates slightly from the experimental spectra (Fig. 4a) on the higher energy side, which may arise from a pump-induced shift of the continuum band.

We find a linear relationship between $\delta E$ and pump intensity (Fig. 4b) as predicted by equation (1). Considering that $\Delta$ and $\langle F\rangle^2$ vary with different experimental conditions, $\mu_{0X}$ is the only intrinsic parameter determining the energy shift. In our experiment, $\mu_{0X}$ is determined as 46 D, which is larger than the reported values of semiconductor quantum wells (about 6 D)[2], carbon nanotubes (about 12 D)[21] and WSe$_2$ monolayers (about 7 D)[20], and is comparable to that in WS$_2$ monolayers (about 51 D)[19]. Equation (1) also implies that the OSE is strongly associated with the intrinsic properties of the material rather than sample preparation method, which is consistent with the observations of OSE in the samples made from different methods

(Supplementary Figs 8 and 9). In contrast, the TA signal caused by OSE depends on the width of the exciton peak (equation (2)), and thus the OSE-induced TA features may show different widths and amplitudes depending on the sample inhomogeneity. It is remarkable that such sharp excitonic features, necessary to observe the OSE, are present in a room-temperature solution-processed semiconductor.

In summary, a large OSE is observed in MAPbI$_3$ at room temperature that is easily resolved. Our observation is consistent with a narrow excitonic transition that is optically active at the absorption bandedge within these inorganic/organic hybrid semi-conductors. We also demonstrate that the OSE-induced absorption change can be controlled by the photon spin polarization, suggesting a method to control spin states which is a requisite for spin-based quantum computing. Realizing a functional spin-based qubit requires the spin coherence time (around 2 ps in this case) to be lengthened so that multiple independent operations[8,10] can be performed before decoherence. The mechanisms limiting the dephasing time and to what extent these can be mitigated in the lead halide perovskite materials remain open questions. It is nonetheless clear from our demonstration of strong OSE in this solution-processed bulk semiconductor that additional experimental study to determine their potential for quantum information applications and femtosecond lasers is warranted.

## Methods

**Synthesis.** The samples were prepared following published literature[33]. A nonstoichiometric mixture of methylammonium iodide and lead iodide (molar ratio of MAI:PbI$_2$ is 1.2:1) was dissolved in a solvent comprising a mixture of N-methyl-2-pyrrolidinone/γ-butyrolactone (7:3, weight ratio) to form an ~50 wt% precursor solution. The precursors were used for the deposition of perovskite films on quartz substrates. Briefly, 50 wt% precursor with 20% more organic salt was dispersed on top of substrate by spin-coating at 4,500 r.p.m. for 25 s, and as-formed wet film was immediately transferred into a diethyl ether (Fisher Chemical) bath to form perovskite film. The perovskite film was further annealed at 150 °C for 15 min with a Petri dish covered to remove excess organic salt.

**Transient absorption spectroscopy.** The femtosecond transient absorption spectrometer is based on a regeneratively amplified Ti:sapphire laser system. The wavelength of the fundamental beam is at 800 nm, and the pulse repetition rate is 1 kHz. The fundamental beam is then split into two beams. One beam is attenuated by neutral density filter and chopped at a rate of 500 Hz, which is used as the pump. The other beam is also attenuated and focused into a sapphire crystal

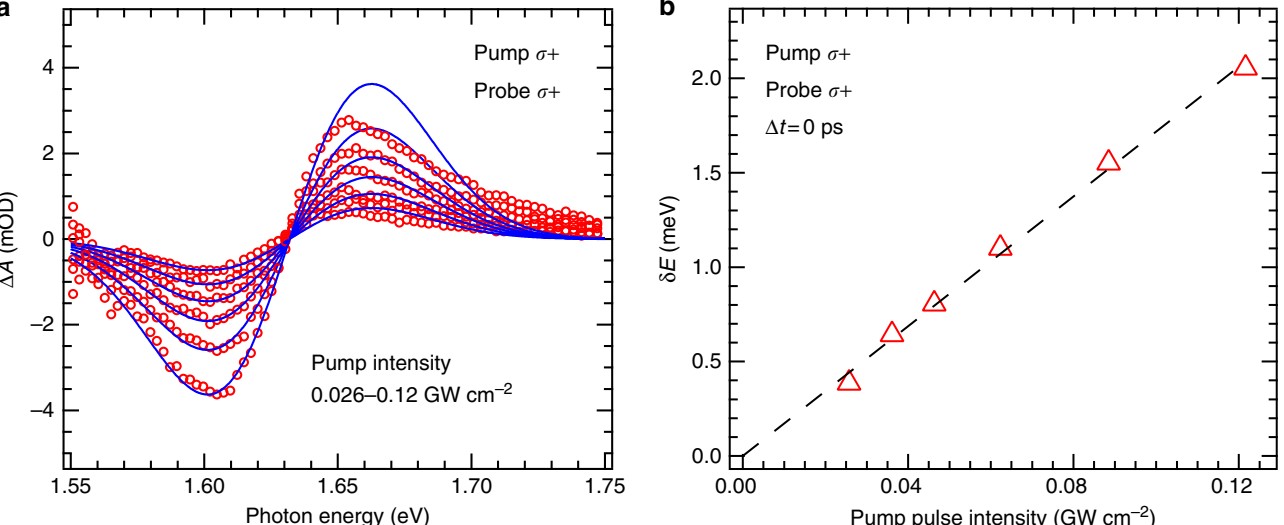

**Figure 4 | TA spectrum fitting and pump intensity-dependent energy shift.** (**a**) OSE-induced TA spectrum (red circles) for different pump intensities recorded at time zero. The blue traces are best fits to equation (2). (**b**) Shift of the exciton transition energy for different pump pulse intensities. The dashed line represents a linear fit of the data points. The average pump intensities are also listed in Supplementary Table 2.

to generate the broadband probe (420–830 nm). The probe is delayed in time with respect to the pump using a motorized translation stage mounted with a retro-reflecting mirror. The pump and probe are spatially overlapped into the sample, and the transmitted probe pulses are directed to the detectors. The polarization of the pump and probe is controlled by polarizers and waveplates.

The total pump intensity is determined by measuring the photon flux after a pinhole with radius of 400 mm at the sample position.

**Data availability.** The authors declare that all data supporting this work are contained in graphics displayed in the main text or in supplemental information. Data used to generate these graphics are available from the authors on request.

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

## Acknowledgements

This work was supported by the Division of Chemical Sciences, Geosciences, and Biosciences, Office of Basic Energy Sciences of the US Department of Energy through the Solar Photochemistry programme under contract DE-AC36-08GO28308 to the National Renewable Energy Laboratory, Golden, Colorado. Perovskite films were supplied from the Hybrid Perovskite Solar Cell program of the National Center for Photovoltaics funded by the US Department of Energy, Office of Energy Efficiency and Renewable Energy, Solar Energy Technologies Office.

## Author contributions

Y.Y. and M.C.B. conceived the original ideas and designed the experiment; Y.Y. carried out the experiment and analysed the data; M.Y. and K.Z. prepared and characterized the samples; Y.Y., J.C.J, J.J.B., J.v.d.L. and M.C.B. wrote the manuscript. All authors discussed the results and commented on the manuscript.

## Additional information

**Competing financial interests:** The authors declare no competing financial interests.

