## [Peer Review File · Nature Communications]

Reviewers' comments

Reviewer #1 (Remarks to the Author):

The manuscript in its revised version well takes into account the reviewer's suggestions, resulting improved if compared to the previous submission. In addition the argumentations in the response to the reviewers letter and the additional information reported in supporting information are appreciable and useful. The work would represent a progress foreseeing optoelectronic exploitation of metal halide perovskite. For these reasons it would suit the elevated standard for publication on Nature Communications.

Reviewer #2 (Remarks to the Author):

The authors have answered to most of the comments raised in the previous referee report. However, I still believe the present results will have a limited impact in the field of quantum information due to the very short spin lifetime (standard quantum error-correction schemes required $\sim 10^4$ operations within the coherence time which is too far from what has been presented by the authors in the present work). The observation of OSE at room temperature and in bulk material is, however, an important experimental result that can generate interest in many other fields. The authors should reconsider the field of application and include in the introduction applications like all optical switching (PRL, 2015, 114, 036802), single photon emission based on perovskite quantum dots for quantum communication (ACS Nano, 2015, 9, pp 10386-10393, ACS Nano, 2016, 10, pp 2485-2490) or ultrafast mode - locked laser (Nature Photonics 2009, 3, pp 729-731).

I can recommend the publication after these minor revisions.

Response to Reviewers

Reviewer #1 (Remarks to the Author):

The manuscript in its revised version well takes into account the reviewer's suggestions, resulting improved if compared to the previous submission. In addition the argumentations in the response to the reviewers letter and the additional information reported in supporting information are appreciable and useful. The work would represent a progress foreseeing optoelectronic exploitation of metal halide perovskite. For these reasons it would suit the elevated standard for publication on Nature Communications.

We thank the reviewer for helpful comments and improving our manuscript.

Reviewer #2 (Remarks to the Author):

The authors have answered to most of the comments raised in the previous referee report. However, I still believe the present results will have a limited impact in the field of quantum information due to the very short spin lifetime (standard quantum error-correction schemes required $\sim 10^4$ operations within the coherence time which is too far from what has been presented by the authors in the present work). The observation of OSE at room temperature and in bulk material is, however, an important experimental result that can generate interest in many other fields. The authors should reconsider the field of application and include in the introduction applications like all optical switching (PRL, 2015, 114, 036802), single photon emission based on perovskite quantum dots for quantum communication (ACS Nano, 2015, 9, pp 10386-10393, ACS Nano, 2016, 10, pp 2485-2490) or ultrafast mode - locked laser (Nature Photonics 2009, 3, pp 729-731).

I can recommend the publication after these minor revisions.

We thank the reviewer for the helpful comments. We have incorporated the suggestions from the reviewer and discussed other potential applications of the optical stark effect both in the introduction of our manuscript and the conclusion.